# Viral–Bacterial Interactions That Impact Viral Thermostability and Transmission

**DOI:** 10.3390/v15122415

**Published:** 2023-12-13

**Authors:** Lorimar Robledo Gonzalez, Rachel P. Tat, Justin C. Greaves, Christopher M. Robinson

**Affiliations:** 1Department of Microbiology and Immunology, Indiana University School of Medicine, Indianapolis, IN 46202, USA; lrobledo@iu.edu (L.R.G.); rtat@iu.edu (R.P.T.); 2Department of Environmental and Occupational Health, School of Public Health, Indiana University, Bloomington, IN 47408, USA; jcgreave@iu.edu

**Keywords:** bacteria–virus interactions, enteric virus, viral transmission

## Abstract

Enteric viruses are significant human pathogens that commonly cause foodborne illnesses worldwide. These viruses initiate infection in the gastrointestinal tract, home to a diverse population of intestinal bacteria. In a novel paradigm, data indicate that enteric viruses utilize intestinal bacteria to promote viral replication and pathogenesis. While mechanisms underlying these observations are not fully understood, data suggest that some enteric viruses bind directly to bacteria, stabilizing the virion to retain infectivity. Here, we discuss the current knowledge of these viral–bacterial interactions and examine the impact of these interactions on viral transmission.

## 1. Introduction

Diarrheal diseases are among the most common causes of death globally, especially in children under the age of five. Enteric viruses are a major type of microbial pathogen that account for diarrheal disease, which kills around 525,000 children each year [1]. These viruses are transmitted through contaminated food, fomites, and water sources and are shed at high levels in the feces of infected individuals. Therefore, enteric viruses pose significant health risks to developed and underdeveloped countries. Additionally, these viruses have a substantial impact on the global economy. For example, the global economic burden of norovirus alone, an enteric virus that is the leading cause of acute gastroenteritis, is estimated to be nearly $65 billion [2]. Unfortunately, despite these burdens, few vaccines or treatments are available for enteric viral infections. Therefore, it is vital to identify potential therapeutic targets to help reduce the burden of disease. Enteric viruses initiate infection in the gastrointestinal tract. During infection in the intestine, these viruses encounter a diverse population of intestinal bacteria. Approximately 10^14^ bacteria, made up of over 500 different species, reside in the ecological niche of the GI tract [3]. Importantly, these symbiotic microorganisms are integral to host health, digestion, and development. However, in contrast to being essential for human health, data also indicate intestinal bacteria can promote the replication and pathogenesis of many pathogenic enteric viruses [4,5,6,7]. While the mechanisms underlying these observations remain unclear, data suggest a conserved mechanism for these interactions is to stabilize the virion to retain viral infectivity [8,9,10,11,12]. The ability to maintain infectivity may play a role during infection in the intestine but is also likely a critical mediator of environmental stability required for viral transmission from host to host. Therefore, since enteric viruses continue to cause outbreaks globally due to ease of transmission, there is a critical need to understand the bacterial–viral interactions that influence environmental stability. Here, we will highlight our latest understanding of these interactions and discuss the consequences. Further, we will discuss how these interactions have real-world implications for environmental transmission.

## 2. Intestinal Bacteria Promote Viral Replication and Pathogenesis

Much of our current understanding of bacterial–viral interactions has been derived from animal models with a specific focus on enteric viruses, given their tropism for the intestine. The intestine is home to a vast community of bacteria that comprise the intestinal microbiome. Based on the sheer number of bacteria in the intestine, it is no surprise that interactions occur between enteric viruses and commensal bacteria, impacting viral infections. Using antibiotic treatments, it was initially hypothesized that bacteria would provide a physical barrier or stimulate an immune response, limiting viral infection in the intestine. The data, however, suggest otherwise. Rather than a barrier against infection, bacteria can promote infection with many enteric viruses. The first examples were poliovirus, reovirus, and mouse mammary tumor virus. First, using an antibiotic approach in mice, the replication of poliovirus and reovirus, enteric viruses from the *Picronaviridae* and *Sedoreoviridae* families, respectively, were significantly reduced when bacteria were depleted [4]. Similarly, the persistence of mouse mammary tumor virus (MMTV), an enteric virus from the *Retroviridae* family, was abolished in the absence of bacteria [5]. Since these initial findings, other enteric viruses, such as norovirus and Coxsackievirus, have also been shown to utilize bacteria. Further, these interactions likely extended out of the gut, as recent data indicate that respiratory bacteria can also enhance influenza A virus [13]. Overall, these initial findings provided a novel paradigm suggesting that bacteria can promote the infection of multiple viruses.

## 3. Bacteria Can Bind Directly to Viruses

Bacteria may have indirect or direct interactions with viruses to promote viral infection. Interestingly, the direct binding of bacteria and/or bacterial cell wall components has been shown to play an essential role in the mechanism of this interaction. Electron microscopy (EM) analysis of this interaction revealed that multiple poliovirus virions bind on the surface of *Escherichia coli* [14]. Similarly, our group recently demonstrated that Coxsackievirus B3 (CVB3), an enteric virus in the Picornavirus family, can bind to *Salmonella enterica* and *E. coli* [10]. The data from our laboratory and others indicate that bacterial cell wall components from both Gram-positive and Gram-negative bacteria may dictate the interaction. Lipopolysaccharide (LPS) and peptidoglycan, the main constituents of Gram-positive and Gram-negative bacteria cell walls, respectively, have been shown to enhance the infectivity of multiple enteric viruses [8,10,11,12]. Further, biotinylated LPS was found to bind directly to the poliovirus capsid [4,8]. This interaction is not limited to viruses from the Picornavirus family, as data indicate that MMTV can incorporate LPS-binding proteins, such as MD-2 and CD14, into the viral envelope during egress. These LPS-binding proteins, embedded in the viral membrane, can bind to bacterial LPS [15]. This interaction facilitates the transfer of LPS to TLR-4 and can help trigger LPS-induced signaling pathways to induce immune evasion pathways [16]. Finally, it is important to note that other components of bacteria can mediate binding to viruses. In contrast to interactions with LPS, studies indicate that rotavirus and norovirus interact with polymorphic human histo-blood group antigens (HBGAs) [17,18]. HBGAs are expressed by the intestinal epithelial cells but are also found in certain bacteria. In vitro and EM data indicate that human norovirus can bind directly to HBGA-expressing bacteria [6,19]. However, norovirus is not limited to binding to HBGA-expressing bacteria, as human and murine norovirus can also bind to various bacteria, including gut commensal bacteria [20,21,22]. Therefore, many viruses have evolved mechanisms to bind to bacteria in distinct ways to influence their replication.

## 4. Bacteria Can Promote the Thermostability of Viruses

Although the outcome of these interactions may vary from virus to virus, one conserved effect in the binding of viruses directly to bacteria is enhanced viral thermostability. Previous studies showed poliovirus survival at high temperatures (42 °C) and enhanced sensitivity to bleach when incubated with LPS [8]. Further, using a cell-free approach, LPS limited the premature release of poliovirus RNA, suggesting that bacteria may help retain infectivity in the environment. In addition to poliovirus, bacteria enhanced the stability of a panel of picornaviruses. Three genera of picornaviruses, including Enterovirus (poliovirus, Coxsackievirus, and echovirus), Kobuvirus (Aichi virus), and Cardiovirus (mengovirus), all displayed an increase in thermostability when incubated with bacteria or bacterial components [12,23]. Further, Gram-positive bacteria and bacterial cell wall components were shown to promote reovirus and murine norovirus thermostability [11,22]. In addition, human norovirus binding to Gram-negative HBGA-expressing bacteria allows for protection from acute stress [24,25]. Overall, these data indicate that promoting viral thermostability may be a conserved mechanism for the direct interactions between intestinal bacteria and enteric viruses.

While the specific bacterial structures required to interact with enteric viruses are still being examined, some clues have been identified as to how this interaction occurs on bacteria. First, LPS and peptidoglycan contain n-acetylglucosamine (GlcNAc), and polysaccharides containing at least six components of GlcNAc were found to be required to promote poliovirus infectivity [8]. Second, using a glycan array, Lu et al. found that acetylated glycans were needed to bind to poliovirus, and while short-chain glycans could bind, GlcNac polymers (>20 units) were required to stabilize the virus. Based on the size requirement, these data suggest that the longer oligomers may bind to multiple sites on the viral capsid and help retain the rigidity to stabilize the virion. Future work is required to examine this hypothesis.

Interestingly, the glycan-binding specificity may not be conserved among enteric viruses. Our laboratory has determined that LPS from *S. enterica*, but not *E. coli*, could promote Coxsackievirus B3 (CVB3) stability by reducing viral decay over time [10]. This differs from poliovirus, which can utilize both *S. enterica* and *E. coli* LPS to retain infectivity. Interestingly, LPS from rough strains of *S. enterica* could not stabilize CVB3. Similar data were observed in poliovirus with rough strains of *E. coli* [8,10]. Since rough LPS comprises the lipid-A moiety and inner core but lacks the O-antigen, this suggests that polysaccharides located in the O-antigen play an essential role in the viral binding and stability of the virion (Figure 1A). Since bacteria have different O-antigen structures, it is unsurprising that differences in these polysaccharides may lead to unique interactions with other enteric viruses [6,18,26]. While the differences in binding bacterial LPS between CVB3 and poliovirus have been identified in vitro, the implication of these distinct interactions is likely why CVB3 is more sensitive to microbiota perturbation in the intestine than poliovirus [9].

These distinct interactions between bacteria and viruses extend outside the Picornavirus family. Like CVB3 and poliovirus, LPS and peptidoglycan can enhance Type 1 Lang (T1L) and Type 3 Dearing (T3D) reovirus strains. However, lipoteichoic acid, a surface molecule from Gram-positive bacteria, and chitin only enhanced T3D thermostability but not the T1L strain [11]. Therefore, specific reovirus strains may have different affinities for distinct bacterial components. Similarly, Budicini and Pfeiffer found that several Gram-positive bacteria enhanced murine norovirus stability at 42 °C, in contrast to Gram-negative bacteria that did not improve viral stability [22]. Interestingly, though, lipoteichoic acid and LPS could stabilize murine norovirus, suggesting that multiple bacterial components from both Gram-positive and Gram-negative bacteria may be sufficient for stabilization. Finally, bacteria can also promote the environmental stability of influenza in a bacterial species-specific manner [13]. In contrast to *Staphylococcus aureus* and *Staphylococcus epidermis*, *Streptococcus pneumoniae* and *Moraxella catarrhalis* enhanced influenza A virus stability. Overall, these data suggest that while the outcome of the interaction may be conserved, the requirements of the components of bacteria that influence the interaction may differ amongst individual viruses.

While some data indicate the requirements of bacteria to interact with enteric viruses, the binding site on virions remains to be determined. Poliovirus and CVB3 are both enteroviruses with capsids comprising four proteins: VP1, VP2, VP3, and VP4. A pentamer of VP1 forms the five-fold axis, while VP2 and VP3 surround the two- and three-fold axes. For picornaviruses, the interaction will likely occur around the five-fold axis based on a few pieces of evidence. First, at the five-fold axis, a depression in the virion surface is known as the viral “canyon” (Figure 1B). Within this depression, the binding site for the primary viral receptor is located, and the base of the canyon contains a hydrophobic pocket occupied by a fatty acid-like ligand known as the “pocket factor”. Previous data have indicated that this pocket factor may function to regulate the thermostability [27,28]. Second, Coxsackievirus A24 has been shown to utilize shallow groves on the virion surface to facilitate carbohydrate binding, such as sialic acid [29]. These groves are found on the five-fold axis and form a positively charged binding site created by the BC- and DE-loop of the VP1 protein. In support of this site being a key player in binding to bacteria, a mutation at residue 99 in VP1, located on the BC loop, impacted the binding of LPS to poliovirus [8]. Finally, another clue in the virion site may come from the bacterial interactions between bacteriophages and bacteria. The tailless single-stranded DNA bacteriophage, ΦX174, utilizes LPS as a primary receptor to infect bacteria. Using cryogenic electron microscopy (cryo-EM) to resolve this interaction revealed that ΦX174 binds to *Salmonella* LPS via positively charged amino acids on the virion near the five-fold axis [30]. Interestingly, the superposition of the crystal structure from CVB3 with ΦX174 revealed conserved channel-like structures along the fivefold axis [31]. Therefore, conserved structures between bacteriophages and enteric viruses may provide clues to the nature of this interaction; however, future structural studies are required to identify these binding sites on eukaryotic viruses.

The implications of enhanced thermostability are currently being investigated. It has been hypothesized that bacteria increase the particle-to-plaque forming unit (PFU) ratio, which promotes infectivity. Since many virions in a viral population are rendered non-infective in the environment due to viral decay, bacteria may limit this by promoting stability. Using cell-free approaches, the data support this hypothesis as LPS can limit the premature release of poliovirus and CVB5 RNA and likely limit the number of empty capsid virions [8,23] (Figure 2). Moreover, in an elegant experiment, an in vivo competition assay using a poliovirus mutant deficient in binding LPS has also provided clues to the importance of this interaction in the environment. In this experiment, mice were orally inoculated with a 1:1 mixture of an LPS-binding deficient mutant poliovirus and wild-type poliovirus. After the mice were infected, the viruses were excreted into the feces, exposing them to the environment. The fecal virus was collected, passaged multiple times in mice, and the ratio of mutant to wild-type virus was determined. Even though both the mutant and wild-type virus could replicate equally in mice, the researchers found less mutant LPS-deficient poliovirus than wild-type virus in the feces over the passages. These data indicate that the mutant virus that could not bind to LPS had reduced fitness in vivo due to the loss in environmental stability [8]. These data indicate that bacteria enhance environmental fitness and suggest there are real-world consequences of bacteria and virus interactions that impact viral transmission.

## 5. Wastewater and Transmission

Viruses can bind to larger particles and microorganisms found in wastewater [32,33,34]. This is not limited to wastewater as this phenomenon is also observed for natural particles and microorganisms found in freshwater and other environmental waters [34,35]. Larger particles in water that viruses may associate with are organic matter, algae, or inorganic matter. These large particles are highly associated with various viruses, including adenovirus, human polyomavirus, and norovirus [32,36,37]. Intestinal bacteria also comprise a considerable portion of municipal wastewater and wastewater-contaminated environments. Hence, it is important to understand the role of bacteria in extending the transmission of viruses in water.

Heat, chlorine treatments, and UV inactivation are common ways viruses can be cleared from wastewater. However, by interacting with bacteria, data indicate that enteric viruses can persist for extended periods even following these environmental factors and treatment procedures [38,39]. For example, Coxsackievirus, echovirus, and poliovirus remain infectious for extended periods when exposed to higher temperatures following incubation with gut bacteria or bacterial components [23]. Additionally, poliovirus, Coxsackievirus, and mengovirus were also shown to have higher resistance to chlorine disinfection when in the presence of bacteria [12]. Further, though viruses can be UV-inactivated through secondary processes through which sensitizers are formed from organic compounds already present in water, viruses have still been shown to be shielded by bacteria from sunlight [40,41]. Thus, these data indicate that bacterial–viral interactions can impact heat treatment, chlorine treatment, and UV inactivation and demonstrate the importance of studying viral–bacterial interactions in the environment.

Enteric viruses can also directly bind to bacteria, which may also affect viral transport in environmental waters. A single particle virus can generally be transported at long distances without settling in water systems for clearance, thereby putting populations over long distances at risk of infection [42]. Understanding this interaction with bacteria and the dynamics of the bacteria can help with understanding viral behavior in flowing waters and allow for more effective treatment measures. For example, viruses that bind to larger bacteria may enable viruses to settle faster in a water system and wastewater treatment plant, making treatment more efficient. More studies are required to demonstrate these interactions in water systems.

Viral–bacterial association may also impact how viruses attach to surfaces, such as human skin, river bedrock, or clothing. Previous studies have extensively explored the transfer of viruses from liquid to skin [43,44]. However, these data assume a free-flowing virus, which differs from most viruses in environmental waters [32]. Therefore, more studies are needed to examine the potential for bacteria to promote viral attachment to various surfaces. We hypothesize that this interaction with bacteria could increase virus prevalence on some surfaces, ultimately impacting transmission [44,45,46]. Moreover, biofilms are the predominant mode by which bacteria exist in the environment. It is intriguing to speculate on how biofilms could either promote or hinder how viruses persist, attach to surfaces, and ultimately impact transmission.

Extending beyond their behavior in water, enteric viruses have also been detected in several other environmental media, such as air, soil, and food [47,48,49]. Recent studies have shown high concentrations of enteric viruses and bacteria are found in aerosols near open wastewater canals [48,50]. Enteric pathogens such as hepatitis A and E virus are also commonly associated with the recall of many food items [49,51]. Moreover, studies have already shown that viral interactions with bacteria significantly alter viral stability in food [52]. Overall, the potential for viruses to interact with bacteria in these different environmental media creates new avenues of transmission; hence, we need to understand how this interplay between enteric viruses and bacteria may impact the risk of infection and transmission. 

## 6. Conclusions and Future Directions

Enteric viruses represent a significant human pathogen. Since enteric viruses continue to cause outbreaks globally due to ease of transmission, evaluating the impact of bacterial–viral interactions on viral stability in the environment is critical. While in vivo data indicate that antibiotics can inhibit enteric viruses, as a clinical approach, it is important to note that the risks of disrupting the effects of the microbiome far outweigh the benefits of limiting viral replication. Intestinal bacteria are critical to multiple vital human functions, including metabolism and developing a healthy immune system. A sledgehammer approach using antibiotics to restrict viral replication would likely cause more unintentional adverse events than benefits. First, it is still being determined if all enteric viruses benefit from interactions with intestinal bacteria. Bacteria may inhibit infection of some enteric viruses, such as rotavirus. Multiple reports indicate that commensal bacteria and probiotics can protect against rotavirus infection [53,54,55,56]. Second, eliminating bacteria may impact viral infections at other sites through indirect interactions with the host immune response. For example, data suggest that immune responses to influenza in the respiratory tract are diminished when commensal bacteria are disturbed by oral antibiotics [57]. Therefore, more targeted approaches are needed to disrupt the interactions between bacteria and these viruses to be viable as a therapeutic. Further, taking advantage of how commensal bacteria stimulate the intestinal immune response may also represent a therapeutic angle that needs to be explored.

Much of the work on virus–bacteria interactions has focused on bacterial interactions with enteric viruses, given their proximity to the vast microbiome in the gut. However, other data suggest that bacteria may impact other viruses at different sites. As previously mentioned, influenza is affected by interactions with bacteria. It has been shown that mice exposed to *Streptococcus pneumoniae* prior to influenza A do not maintain a robust germinal center B cell response, suggesting that coinfection or past pathogen exposure affects the generation of antiviral antibodies [58]. Another study reported that *Staphylococcus aureus* secreted protein lipase 1 which enhances influenza A replication in vitro and that this pro-viral effect is maintained during in vivo infection [59]. Multiple studies have also shown an association between SARS-CoV-2 infection and the respiratory tract microbiome. There is greater dysbiosis in the respiratory microbiota in COVID-19 patients, with lower bacterial diversity when compared to healthy patients [60,61,62]. These findings suggest that non-enteric viruses and bacteria have a relationship, and both contribute to the outcome of viral infection. More work must be carried out to identify both direct and indirect effects of these interactions on pathogenicity and disease progression.

The interplay between these viruses and bacteria presents a significant biological challenge since these interactions are likely crucial in understanding the risks to the public and limiting future outbreaks. To meet this challenge, future studies must examine the interaction from both the bacteria and virus perspective. For example, there is a delicate balance between stabilizing the virion to limit viral decay and the metastable interactions that must occur once the virus interacts with the host receptor to release viral RNA into the cell. Therapeutics have been derived to make viruses more stable yet less infective, so how viruses walk this tightrope to utilize bacteria for thermostability needs to be addressed. Second, most current studies reflect on the benefits of this interaction on viral infection; however, bacteria may be altered by binding to these viruses. For example, astroviruses can disrupt the gut microbiome in humans, and avian strains can lead to atypical outgrowth of *E. coli* in poultry [63,64,65]. Moreover, it is unclear if bacteria may also benefit from interacting directly with viruses. It is possible that bacteria may utilize enteric viruses to hide from the immune response or gain additional unknown fitness benefits. Finally, since the intestine is home to many different bacterial species that localize to different areas of the gut, it is intriguing to speculate how this may impact tissue tropism for different enteric viruses. Enteric viruses may use binding to specific bacteria as a guide to direct the initial infection of a select subset of intestinal cells. With these exciting questions still needing to be addressed, we anticipate a new growth of knowledge within this new field of virology in the coming years. Further, understanding how various viruses interact with bacteria will help develop novel therapeutics and prevent future enteric virus outbreaks.

## Figures and Tables

**Figure 1 viruses-15-02415-f001:**
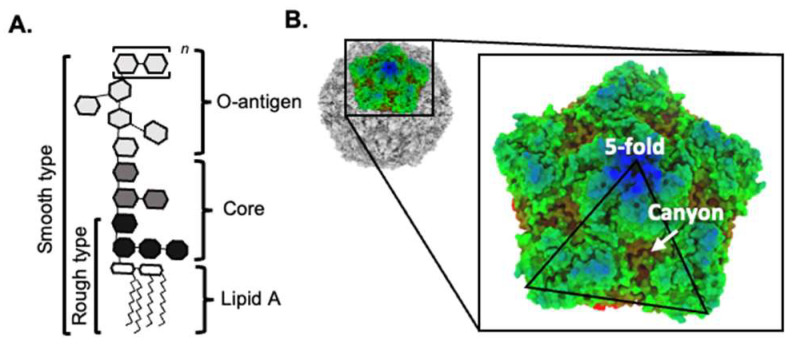
Structure of bacterial LPS and enterovirus capsid. (**A**) The basic structure of bacterial lipopolysaccharide. The O-antigen can have repeating oligosaccharide units (designated *n*). (**B**) The topography of the CVB3 capsid (RCSB PDB: 1COV), including the 5-fold axis and the canyon.

**Figure 2 viruses-15-02415-f002:**
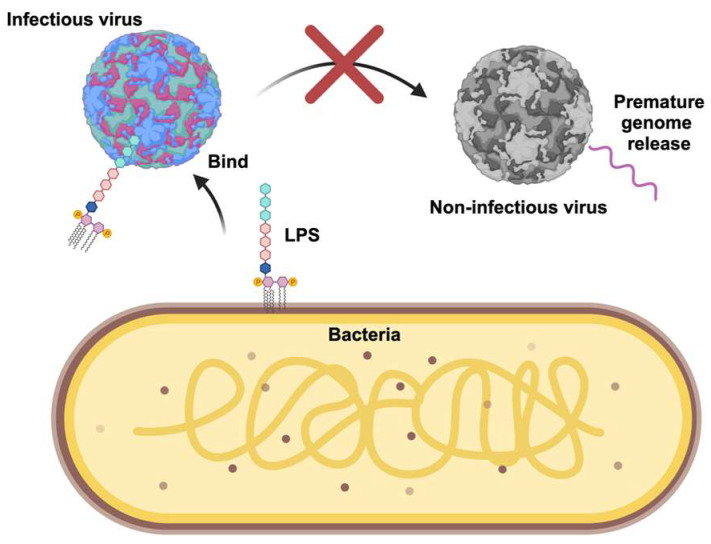
Proposed model for how bacteria may limit inactivation of enteric viruses. Bacteria and bacterial LPS can bind directly to the viral capsid and stabilize the virus, limiting premature genome release. Figure created using Biorender.com.

## Data Availability

Data sharing not applicable.

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
