# Peer review of "Viral–Bacterial Interactions That Impact Viral Thermostability and Transmission"

_viruses, 2023, doi:10.3390/v15122415_

Round 1
Reviewer 1 Report
Comments and Suggestions for Authors
This review is clear and well documented. I have a few minor comments/suggestions:
1) Lines 108-109: Evidences that bacteria promote thermostability for human norovirus as well were provided by Li D et al, Front in Microbiol 2015 and Xu Z et al, Front in Microbiol 2021. Although Budiccini et al found that only Gram+ bacteria could protect MNV, Li et al and Xu et al found Gram- bacteria to be protective for NoV. However, in these two studies only the capsid or genome integrity rather than virus infectivity could be assessed.
2) Lines 165-167: It is written 5-fold access instead of axis
3) Lines 195-197: The sentence does not seem very clear to me… I think it should be “…when mixed with wild type virus, the ratio decreased, indicating a fitness cost due to a loss of environmental stability.”

Reviewer 2 Report
Comments and Suggestions for Authors
This manuscript discusses current understanding of virus-bacterial interaction that potentially impact viral thermostability and transmission. It is very interesting topic and nicely written. This reviewer has one comment, but this could be accepted in the current form.
Comment:
Although I understand that this manuscript focuses on interaction between intestinal bacteria and enteric viruses, and the manuscript has few sentences regarding influenza virus, I wonder if other non-enteric viruses also have benefit by interacting bacteria, which alter the outcome. For example, do viral-bacterial interaction enhance replication, pathogenicity, and/or transmission of highly pathogenic respiratory viruses (e.g., SARS-CoV-2, highly pathogenic influenza virus) in animal model, or enhance persistence or latency of particular viruses such as HIV, HCV, HBV?
Reviewer 3 Report
Comments and Suggestions for Authors
Enteric viruses using binding to specific bacteria as a guide to direct the initial infection of a select subset of intestinal cells could be a new growth of knowledge and a new field of virology. It could be an excellent new challenge to discover also for novel therapeutics plans.
Very interesting
